# Improved Quality of Life Following Addiction Treatment Is Associated with Reductions in Substance Use

**DOI:** 10.3390/jcm8091407

**Published:** 2019-09-06

**Authors:** Victoria Manning, Joshua B. B. Garfield, Tina Lam, Steve Allsop, Lynda Berends, David Best, Penny Buykx, Robin Room, Dan I. Lubman

**Affiliations:** 1Monash Addiction Research Centre, Eastern Health Clinical School, Monash University, 110 Church Street, Richmond, Victoria 3121, Australia; 2Turning Point, Eastern Health, 110 Church Street, Richmond, Victoria 3121, Australia; 3National Drug Research Institute, Curtin University, 2 Building 609, Curtin University, 7 Parker Place, Bentley 6102, Australia; 4TRACE Research, 1/209 Nicholson St, Footscray, Victoria 3011, Australia; 5National Drug and Alcohol Research Centre, University of New South Wales, Sydney 2052, Australia; 6Department of Law, Criminology, and Community Justice, Sheffield Hallam University, Howard St, Sheffield S1 1WB, UK; 7School of Humanities and Social Science, University of Newcastle, University Drive, Callaghan 2308, Australia; 8Centre for Alcohol Policy Research, La Trobe University, Bundoora, Victoria 3086 Australia; 9Centre for Social Research on Alcohol & Drugs, Department of Public Health Sciences, Stockholm University, SE - 106 91 Stockholm, Sweden

**Keywords:** quality of life, substance use treatment, substance use disorder, reduced substance use, abstinence, treatment outcome, addiction, alcohol and drugs

## Abstract

People seeking treatment for substance use disorders (SUD) ultimately aspire to improve their quality of life (QOL) through reducing or ceasing their substance use, however the association between these treatment outcomes has received scant research attention. In a prospective, multi-site treatment outcome study (‘Patient Pathways’), we recruited 796 clients within one month of intake from 21 publicly funded addiction treatment services in two Australian states, 555 (70%) of whom were followed-up 12 months later. We measured QOL at baseline and follow-up using the WHOQOL-BREF (physical, psychological, social and environmental domains) and determined rates of “SUD treatment success” (past-month abstinence or a statistically reliable reduction in substance use) at follow-up. Mixed effects linear regression analyses indicated that people who achieved SUD treatment success also achieved significantly greater improvements in QOL, relative to treatment non-responders (all four domains *p* < 0.001). Paired *t*-tests indicated that non-responders significantly improved their social (*p* = 0.007) and environmental (*p* = 0.033) QOL; however, their psychological (*p* = 0.088) and physical (*p* = 0.841) QOL did not significantly improve. The findings indicate that following treatment, QOL improved in at least some domains, but that reduced substance use was associated with both stronger and broader improvements in QOL. Addressing physical and psychological co-morbidities during treatment may facilitate reductions in substance use.

## 1. Introduction

Previous outcome studies have consistently reported high rates of cessation or substantial reduction in substance use following treatment [1,2,3,4,5,6,7,8,9]. However, substance use disorders (SUD) affect multiple other areas of health and psychosocial functioning, and improvement in these domains is also a common goal of treatment. As noted by Ray, Lim, and Shoptaw [10], “A vague but universal objective among individuals entering addiction treatment is to ‘get their life back’”. Indeed, a panel of treatment and research experts convened by the US National Institute on Drug Abuse (NIDA) in 2009 recommended that SUD treatment outcome research seek to measure a number of other domains, specifically, craving, psychosocial functioning, self-efficacy, quality of life (QOL), and social support [11]. However, Tiffany et al. [11] observed that, at that time, addiction research lagged behind other biomedical fields in terms of evaluating QOL outcomes.

Nevertheless, several major treatment outcome studies have included measures of health-related QOL, such as the 12-item [12] or 36-item versions of the Short Form Health Survey [13] (SF-12 and SF-36, respectively). The Australian Treatment Outcome Study (ATOS) of people entering treatment for heroin dependence found improvements in SF-12 physical health, which reached levels similar to general population norms one year following treatment entry, whilst psychological health remained below population norms despite improving [14,15,16]. Similarly, in the UK Drug Treatment Outcomes Research Study (DTORS), changes in SF-12 scores indicated improvements in psychological health 3–5 months after treatment, though again, with mean scores remaining below general population norms, whilst mean physical health scores, already similar to general population norms at baseline, remained unchanged [8]. Other studies have observed improvements in physical and psychological well-being after treatment using measures other than QOL scales [1,3,5,7].

Another widely-used QOL measure is the brief version of the World Health Organization Quality of Life scale (WHOQOL-BREF) [17] which, in addition to physical and mental QOL, also assesses social, and environmental (e.g., financial resources; opportunities for leisure and skill-acquisition) QOL. Significant improvements in all four QOL domains have been reported following commencement of treatment in several studies of OST for opioid use disorder [18,19,20,21] and also at 6- and 12-month follow-ups in a large alcohol use disorder treatment study [22]. In another study focused on women receiving treatment for SUD, all WHOQOL-BREF domains except social QOL significantly improved 6 months after treatment intake [23].

Poor QOL as a result of a substance-using lifestyle is often the impetus for help-seeking. Numerous studies report that negative social consequences of substance use are often major reasons for treatment-seeking [24,25,26,27], with one study identifying that they were a stronger predictor of entering treatment than dependence symptoms [27]. Medical and psychological problems have also been reported as either predictors, or common precipitators, of SUD treatment-seeking in several studies [24,25,27]. The corollary is that improvements in social, psychological, and physical QOL are often likely to be an important outcome for clients, and one which may help them maintain reductions in substance use over the long term. However, it remains unclear whether these two types of outcomes (reduced substance use and improved QOL) are related to or dependent on each other. Few studies have examined these associations using measures specifically designed to assess QOL. In DTORS, those who reported past 3-month abstinence from illicit drugs (other than cannabis) at a 33-month follow-up had nearly five times the odds of an improvement in SF-36 psychological health score greater than one standard deviation (SD) above the total sample’s average improvement [9]. In contrast, Tracy et al. [23] found relatively little evidence that substance use at follow-up was associated with QOL outcomes on the WHOQOL-BREF, with abstinence from alcohol at follow-up and environmental QOL being the only significant association.

Kiluk et al. [28] recently challenged the value of using global measures of psychosocial functioning in addiction treatment research and suggested measures of specific wellbeing/psychosocial consequences that can be causally attributed to substance use are more clinically meaningful. However, it may be difficult to distinguish between consequences that are clearly substance use-related, indirectly related, or unrelated, especially over the long time-frames of many treatment outcome studies. Indeed, the paucity of studies addressing this question precludes us from drawing any conclusion on whether or not QOL is related to other clinically meaningful outcomes. Thus, to advance the literature around the validity of general QOL scales as outcome measures, we analysed data from a large Australian treatment outcome study to examine whether reducing/ceasing substance use was associated with improvements on the four QOL domains measured by the WHOQOL-BREF. We hypothesised that participants who reduced/ceased use of their primary drug of concern (PDOC) would show larger improvements in QOL than those who did not reliably reduce their PDOC use (“treatment non-responders”). Nonetheless, we also explored whether there were still detectable improvements in QOL among treatment non-responders, despite their continued substance use.

## 2. Methods

‘Patient Pathways’ was a prospective treatment outcome study. The participants, measures, and procedures have been described extensively elsewhere [29,30,31] and, for the sake of brevity, only aspects pertinent for understanding the current analyses are summarised below. Ethics approval was provided by the Eastern Health Human Research Ethics Committee (E17/1112), Monash University Human Research Ethics Committee (201200020) and Curtin University (HR11/2012).

Participants and setting: 796 participants aged ≥18 years, who had commenced treatment within the past month, were recruited from 21 different specialist alcohol/drug services across 37 different sites in Victoria and Western Australia (WA) including acute withdrawal, residential rehabilitation and outpatient settings. Baseline interviews were conducted between January 2012 and January 2013 and 555 (70%) participants completed a 1-year follow-up. Due to missing data regarding either substance use or QOL, the present analyses included 536 participants (for physical and psychological QOL) or 535 participants (for social and environmental QOL).

Relevant measures: At both baseline and follow-up, researchers administered a battery of questionnaires which included measures of demographic characteristics, substance use, and QOL. Participants were asked to identify their PDOC and frequency of use of all licit and illicit substances in the past 30 days was measured using the ASSIST [32]. QOL was measured using the 24 items measuring the four domains of the WHOQOL-BREF (physical, psychological, social and environmental). The WHOQOL-BREF has been shown to have cross-cultural applicability and strong reliability and validity in both general population and physical and mental health treatment samples [17], as well as an alcohol use disorder treatment sample [22]. Australian general population norms were published for the WHOQOL-BREF in 2006 [33].

Analyses: SUD treatment success was defined as achieving either of two outcomes: abstinence from the PDOC (the predominant treatment goal for 74.0% of participants) in the 30 days prior to follow-up, or a statistically-reliable reduction in days of PDOC use in the past month at follow-up, relative to baseline. Reliable change criteria (RCC) were calculated using the Jacobson and Truax [34] formula, utilising reliability indices reported by Ryan et al. [35]. Further details of these calculations are provided in Appendix A. Differences in QOL outcomes at follow-up between participants who achieved treatment success and non-responders were analysed using Stata version 14. Mixed effects linear regression was undertaken separately for each WHOQOL-BREF domain, assessing the time × treatment success interaction to test whether participants achieving SUD treatment success showed a greater change in WHOQOL-BREF scores between baseline and follow-up, relative to SUD treatment non-responders. Multivariate analysis adjusted for age, sex, PDOC, primary index treatment (PIT) type at baseline (i.e., outpatient, withdrawal management, or residential rehabilitation), whether or not the PIT was ended early, and time between baseline and follow-up. Separate paired *t*-tests were used to test whether changes in WHOQOL-BREF scores were significant in treatment responders and non-responders. One-sample *t*-tests were used to test whether QOL scores differed significantly from general population means. Independent samples *t*-tests were used to compare SUD treatment responders’ WHOQOL-BREF scores to those of SUD treatment non-responders at baseline and at follow-up.

## 3. Results

Table 1 describes baseline characteristics of the 536 participants included in these analyses. Mean scores on all domains of the WHOQOL-BREF were between 1 to 2 SDs below the Australian general population mean (physical: *t* (535) = −24.03; psychological: *t* (535) = −27.52; social: *t* (534) = 26.85; environmental: *t* (534) = −19.44; all *p* < 0.001). Follow-ups occurred a mean of 381.1 days (SD = 71.9) after the baseline interview, and at the time of the follow-up, 66.1% of participants had completed their PIT, 5.6% were still engaged with it, and 28.2% had left or been expelled from their PIT prior to completion. Half (51.9%) met the criterion for SUD treatment success (38.4% because they were abstinent at follow-up, 13.4% because they reliably reduced number of days of use of the PDOC, despite not being abstinent). Participants missing from analyses due to either loss to follow-up or missing QOL or treatment outcome data (*n* = 260) were significantly younger (mean = 35.00 ± 10.79 versus mean = 38.16 ± 10.64, *t* (789) = −3.897, *p* < 0.001); and more likely to have amphetamines, and less likely to have alcohol, as their PDOC (Pearson *χ*^2^ = 12.784, *p* = 0.012). Participants recruited from residential rehabilitation (41.7%) and acute withdrawal (33.0%) were more likely to be missing from analyses than those recruited from outpatient settings (22.0%, *χ*^2^ = 19.802, *p* < 0.001). There were no differences in gender between those included in and those missing from analyses (*χ*^2^ = 1.553, *p* = 0.213) nor in any domain of baseline WHOQOL-BREF scores (all *p* > 0.05).

Table 2 shows WHOQOL-BREF scores for each QOL domain separately for treatment responders and non-responders. Both groups had similar scores across all four domains at baseline. Among treatment responders, improvements in QOL were significant for all four domains. The increases in score were equivalent to improvements of 0.73, 1.07, 0.92, and 0.75 general population SDs for the physical, psychological, social, and environmental domains, respectively. Thus, at follow-up, their mean scores were all within one SD of population norms (physical: −0.46, SD = 1.19; psychological: −0.77, SD = 1.50; social: −0.72, SD = 1.43; environmental: −0.49, SD = 1.26). Nevertheless, these scores were still significantly lower than general population norms (physical: *t* (277) = −6.45; psychological: *t* (277) = −8.57; social: *t* (277) = −8.35; environmental: *t* (277) = 6.51; all *p* < 0.001). Among treatment responders, non-abstinent participants who reliably reduced their PDOC use showed similar QOL improvements to those who achieved abstinence at follow-up (see Appendix A).

For SUD treatment non-responders, there were no significant changes in physical or psychological QOL scores between baseline and follow-up. While increases in social and environmental QOL were statistically significant in SUD treatment non-responders, the magnitude of improvement was equivalent to only 0.24 and 0.18 Australian general population SDs, respectively. Thus, at follow-up, non-responders’ mean scores were still more than one SD below general population norms for all domains (physical: −1.12, SD = 1.13, *t* (257) = −15.83; psychological: −1.60, SD = 1.54, *t* (257) = −16.63; social: −1.21, SD = 1.35, *t* (257) = −14.38; environmental: −1.03, SD = 1.38, *t* (257) = −11.94; all *p* < 0.001). As shown in Table 2, linear regression analyses found that the time *x* treatment success interactions were significant for all QOL domains, confirming that improvements in QOL were significantly larger in treatment responders than in non-responders.

Further exploratory repeated measures ANOVA analyses examined whether these interactions were consistent across PDOC (i.e., alcohol versus illicit drug). There were no significant three-way interactions between PDOC, time, and treatment success for environmental, social or physical QOL, but there was a significant three-way interaction for psychological QOL (*F* (1,532) = 4.874, *p* = 0.028, η^2^_p_ = 0.009). This interaction is explored further in Appendix A. Importantly, despite this three-way interaction, the two-way interaction between time and treatment success was significant in both PDOC groups (see Appendix A), suggesting that the main finding (i.e., that improvement in psychological QOL was greater in treatment responders than in non-responders) was consistent, despite this association being stronger in those with alcohol as their PDOC. In addition, there were no 2-way interactions between PDOC and time, suggesting no general effect of PDOC on changes in QOL over time.

## 4. Discussion

Consistent with previous findings of improved QOL following engagement with treatment for SUD [8,16,18,19,20,21,22,23], we observed improvements in all four domains of the WHOQOL-BREF at the one-year follow-up. However, for physical and psychological QOL, these improvements were only significant in participants who substantially reduced their frequency of use of, or ceased using, their PDOC. Although SUD treatment non-responders showed significant increases in social and environmental QOL, these were significantly smaller than the increases shown by SUD treatment responders. In contrast to Kiluk et al.’s [28] proposition that outcomes indexed by general measures of well-being were relatively unrelated to substance use outcomes, these findings suggest that outcomes measured by the WHOQOL-BREF were robustly related to substance use outcomes, suggesting that this instrument offers important and relevant information about treatment outcomes.

While some previous reports found associations between improved psychological well-being and substance use outcomes [9,36,37,38], several studies have reported that these associations are weak, and are also generally weak or absent for indicators of physical and social QOL [23,36,37,38,39]. However, of the reports suggesting weak or absent associations between changes in well-being and substance use following treatment, only one [23] used a measure designed specifically to measure QOL, and this was conducted with a modest-sized, female-only sample. Moreover, these previous studies were based in the US, and it is unclear whether differences between the US and Australian treatment systems, or more broadly (e.g., differences in culture or patterns of substance use) may explain why we found much stronger evidence for these associations than these previous studies.

Given that this was a naturalistic non-randomised cohort study with only one follow-up in which QOL and substance use outcomes were assessed at the same time, we cannot draw causal conclusions regarding our findings. Our ability to interpret these findings is also limited by the fact that 30% of the baseline sample was lost to follow-up. Thus, our analyses might overestimate actual improvements in QOL, and this may bias our findings regarding the degree of association between these improvements and substance use outcomes. Moreover, participants missing from these analyses were younger than those included in these analyses, and differed from those included in terms of PDOC and PIT, and this may also bias our analyses and/or limit generalisability. Moreover, improvements in QOL may partly reflect regression to the mean, particularly given that people are likely to be at a particularly “low point” in their lives when entering SUD treatment. Thus, in the absence of a non-treatment control group, the improvements in social and environmental QOL in treatment non-responders cannot reliably be attributed to SUD treatment. While it could be argued that regression to the mean in treatment responders should be reflected in all four QOL domains, not just social and environmental, their lack of a significant improvement in physical and psychological QOL may be due to the fact that these domains are more intrinsically related to substance use. One final limitation is that psychiatric diagnosis was not assessed, and therefore its impact on changes in QOL could not be examined.

Despite these caveats, our findings are suggestive of two similarly plausible explanations (which are not mutually exclusive, and indeed may operate together in a mutually reinforcing manner). One is that substance use (directly and/or indirectly) negatively impacts, and/or prevents improvements in, QOL. Thus, those who reduce their substance use are “freed” from these negative impacts and are more likely to experience improvements in QOL. This would suggest that treatments effective at reducing substance use (i.e., psychotherapeutic interventions, peer support/mutual aid, pharmacotherapy, and residential rehabilitation) are also inherently beneficial to QOL. This emphasises the importance of increasing the availability of, and facilitating clients’ access to, these treatments, as well as finding ways to maximise rates of treatment completion [31]. Nevertheless, the significant (albeit small) improvements in social and environmental QOL seen among treatment non-responders suggest that treatment may be beneficial even to those who achieve little or no change in their substance use. The unexpected improvements in social QOL could be attributed to the fostering of new social relationships, including connecting with others in SUD treatment or mutual aid/peer support (attended by 48.6% [31]), and/or by strengthening social support or pre-existing relationships by virtue of the participant entering treatment to address their SUD. Moreover, by being in treatment participants could have been referred to social and welfare services (e.g., housing, employment agencies etc.), leading to improved financial security, safety, comfort and convenience of their living environment, and improved access to resources and meaningful activities, thereby accounting for an improved environmental QOL.

The alternative explanation is that experiencing improvements in QOL facilitates, or motivates, reductions in substance use. For example, in line with the self-medication hypothesis [40], participants may have continued using substances as a coping mechanism for managing psychological distress, pain, or severe/complex psychosocial issues, and may become much more likely to reduce their substance use only after finding alternate solutions that improve their QOL. This would suggest that, for clients with poor physical and/or psychological wellbeing (a substantial proportion of the treatment-seeking population) and with poor QOL more generally, finding solutions to these problems would often be necessary to achieve reductions in substance use. Indeed, Hunt and Azrin’s [41] small trial of the “community reinforcement” approach, an intensive approach aimed at improving social and environmental QOL, contingent on abstaining from alcohol, found that it led to high rates of abstinence among alcohol-dependent men. This also emphasises the importance of integrated care and referral between addiction treatment and other mental health, medical, social (e.g., housing and employment) services, and peer support, to achieve not only improvements in these domains, but potentially in substance use as well. Nonetheless, the influence of other variables (e.g., demographic and clinical characteristics) on changes in QOL, particularly whether other factors moderate or mediate associations between treatment success and improvement in quality of life, warrants examination in future research.

## Figures and Tables

**Table 1 jcm-08-01407-t001:** Baseline demographic and clinical characteristics of participants.

Variable		Value
Age (years), mean (SD)		38.2 (10.6)
Sex	Male	324 (60.4%)
	Female	210 (39.2%)
	Missing	2 (0.4%)
Primary drug of concern (PDOC)	Alcohol	268 (50.0%)
	Cannabis	84 (15.7%)
	Opiates	81 (15.1%)
	Amphetamines	90 (16.8%)
	Other	13 (2.4%)
Primary index treatment (PIT)	Outpatient	167 (31.2%)
	Acute withdrawal	235 (43.8%)
	Residential rehabilitation	134 (25.0%)
Physical quality of life, mean (SD)	WHOQOL-BREF numerical score	52.4 (20.3)
	Australian population z-score ^1^	−1.16 (1.12)
Psychological quality of life, mean (SD)	WHOQOL-BREF numerical score	45.3 (21.3)
	Australian population z-score ^1^	−1.81 (1.52)
Social quality of life, mean (SD)	WHOQOL-BREF numerical score	43.4 (24.2)
	Australian population z-score ^1^	−1.55 (1.33)
Environmental quality of life, mean (SD)	WHOQOL-BREF numerical score	59.1 (19.0)
	Australian population z-score ^1^	−1.23 (1.46)

WHOQOL-BREF: World Health Organization Quality of Life Scale (Brief version). ^1^ WHOQOL-BREF scores expressed as Australian general population standard deviations from the Australian population mean, according to normative data published by Hawthorne et al. [33].

**Table 2 jcm-08-01407-t002:** Changes in WHOQOL-BREF scores between baseline and follow-up, compared between those who achieved treatment success vs. those who did not, with results of linear regression models testing the time x treatment success interactions.

QOL Domain ^3^	Treatment Success	Baseline Mean (SD)	Follow-up Mean (SD)	*P*-Value for Within-Category Change from Baseline to Follow-up ^1^	Unadjusted			Adjusted ^2^		
					Estimated Difference	95% CI	*p*-Value	Estimated Difference	95% CI	*p*-Value
Physical	No	53.0 (18.5)	53.3 (20.5)	0.841	12.99	9.43–16.56	<0.001	13.22	9.66–16.78	<0.001
	Yes	51.9 (21.8)	65.1 (21.6)	<0.001						
between-groups *p*-value		0.504	<0.001							
Psychological	No	45.8 (19.9)	48.2 (21.6)	0.088	12.65	8.73–16.58	<0.001	12.98	9.06–16.89	<0.001
	Yes	44.8 (22.5)	59.8 (21.0)	<0.001						
between-groups *p*-value		0.588	<0.001							
Social	No	45.2 (23.4)	49.6 (24.5)	0.007	12.37	7.63–17.11	<0.001	12.91	8.20–17.62	<0.001
	Yes	41.7 (24.8)	58.5 (26.0)	<0.001						
between-groups *p*-value		0.101	<0.001							
Environmental	No	59.3 (18.7)	61.7 (18.0)	0.033	7.24	3.84–10.64	<0.001	7.38	3.96–10.80	<0.001
	Yes	59.0 (19.3)	68.7 (16.4)	<0.001						
between-groups *p*-value		0.816	<0.001							

QOL: quality of life. Estimated difference refers to estimated difference in effect between the abstinent group, compared to non-abstinent, group over time. ^1^
*p* value for change from baseline to follow-up in paired-samples *t*-tests conducted separately within each treatment success category. ^2^ Adjusted for age, sex, time between baseline and follow-up, primary index treatment (PIT) type, whether or not PIT was ceased early (i.e., neither completed as planned nor still continuing at follow-up), and primary drug of concern. ^3^ For comparison, Australian general population norms published by Hawthorne et al. are 73.5 (18.1), 70.6 (14.0), 71.5 (18.2), and 75.1 (13.0) for physical, psychological, social, and environmental domains, respectively.

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
