# Peer review of "Improved Quality of Life Following Addiction Treatment Is Associated with Reductions in Substance Use"

_jcm, 2019, doi:10.3390/jcm8091407_

Round 1

Reviewer 1 Report

Thank you for the opportunity to review the brief report titled: Improved Quality of Life Following Addiction Treatment is Associated with Reductions in Substance Use. Overall, this was a well written paper on a topic that I believe is relevant to the journal’s audience and to the broader field of addiction research and treatment. I also believe this brief report makes an important contribution to the literature and that it provides a novel take on Quality of Life (QoL) as an indicator of treatment outcomes. It also makes a sound argument for increasing access to SUD treatment and it argues that improvements in QoL – albeit small – can be detected post-treatment even among those who otherwise are considered treatment non-responders. Below are comments about the manuscripts, as well as a couple of questions that could be addressed by the authors (if needed). The study under review examines four key domains of Quality of Life in a large sample of adults who received treatment for SUD. Data was analyzed appropriately using mixed effects linear regression and paired t-tests. The literature review on QoL is robust and relevant, especially for a Brief Report. The rationale for the study was also well developed. The authors make it clear that this study does not have a control group, so changes at post-treatment cannot be attributed to the treatment. Yet, there are in my opinion some interesting findings that contribute to the discussion about QoL in the context of substance use and treatment response. As expected, results indicate that although treatment responders (past-month abstinence or a statistically reliable reduction in substance use) and treatment non-responders (no significant reduction in substance use) did not differ on indices of QoL at baseline, treatment responders saw greater improvements across all four QoL domains compared to non-responders. Interestingly, compared to baseline, the non-responders experienced significant improvements in their social and environmental QoL, but not their physical and psychological QoL, at 12-month follow-up. Given these changes in two domains, is there a theoretical or practical angle or interpretation as to why social and environmental QoL improves without ceasing use, while psychological and physical domains do not? In other words, the authors note that treatment can be beneficial even if the client does not cease or reduce use (as seen in the improved QoL compared to baseline), but it appears psychological and physical domains – which often are the reason for seeking treatment in the first place – are not as amenable to change. The authors do suggest, as a plausible explanation, that psychological and physical areas of QoL are linked closely to psychological functioning overall, and that clients who have not ceased use are, in fact, self-medicating. In brief, I agree with the community reinforcement approach and need for integrated care, but a bit more on the self-medication hypothesis (in both treatment responders and non-responders) would be helpful. Another question that the authors may want to address in the discussion or as a suggestion for future studies is the extent to which the changes in QoL could be “drug dependent” such that some substances present more of a drain on QoL (and/or psychological well-being) compared to others? The authors do mention the cultural factor and a possible difference between samples in the USA and Australia, but I would also like to see that other descriptive or demographic variables are included in future research. Overall, a well-written paper with a solid rationale, method, and results section.

Author Response

Reviewer 1.

Interestingly, compared to baseline, the non-responders experienced significant improvements in their social and environmental QoL, but not their physical and psychological QoL, at 12-month follow-up. Given these changes in two domains, is there a theoretical or practical angle or interpretation as to why social and environmental QoL improves without ceasing use, while psychological and physical domains do not? In other words, the authors note that treatment can be beneficial even if the client does not cease or reduce use (as seen in the improved QoL compared to baseline), but it appears psychological and physical domains – which often are the reason for seeking treatment in the first place – are not as amenable to change. The authors do suggest, as a plausible explanation, that psychological and physical areas of QoL are linked closely to psychological functioning overall, and that clients who have not ceased use are, in fact, self-medicating. In brief, I agree with the community reinforcement approach and need for integrated care, but a bit more on the self-medication hypothesis (in both treatment responders and non-responders) would be helpful.

We thank the reviewer 1 for their reflections on the strengths of the paper.

This was an unexpected finding, particularly following Tracy et al.’s (2012) finding that improvements in social and environmental QOL follow improvements in physical and psychological QOL. We have added further discussion on page 15 of the amended manuscript (tracked-changes version) to explore possible reasons for this. In particular we added the following sentences to the end of the first paragraph on page 15:

“The unexpected improvements in social QOL could be attributed to the fostering of new social relationships, including connecting with others in SUD treatment or mutual aid/peer support (attended by 48.6% [31]), and/or by strengthening social support or pre-existing relationships by virtue of the participant entering treatment to address their SUD. Moreover, by being in treatment participants could have been referred to social and welfare services (e.g., housing, employment agencies etc.), leading to improved financial security, safety, comfort and convenience of their living environment, improved access to resources and meaningful activities, thereby accounting for an improved environmental QOL.”

In addition, we have amended the first 2 sentences of the following paragraph so that they now say:

“The alternative explanation is that experiencing improvements in QOL facilitates, or motivates, reductions in substance use. For example, in line with the self-medication hypothesis in the dual diagnosis literature [40], participants may have continued using substances as a coping mechanism for psychological distress, pain, or severe/complex psychosocial issues, and may become much more likely to reduce substance use only after experiencing “relief” from these QOL problems.”

Another question that the authors may want to address in the discussion or as a suggestion for future studies is the extent to which the changes in QoL could be “drug dependent” such that some substances present more of a drain on QoL (and/or psychological well-being) compared to others? The authors do mention the cultural factor and a possible difference between samples in the USA and Australia, but I would also like to see that other descriptive or demographic variables are included in future research.

In our analysis of Changes in WHOQOL-BREF scores between baseline and follow-up, comparing treatment success vs. non-responders using linear regression models testing the time x treatment success interactions, we did control for and primary drug of concern. However to explore this further in response to points raised by both reviewers, we ran repeated measures ANOVAs in which primary drug of concern (PDOC) was entered as an additional between-groups variable (dichotomised as alcohol vs. other drugs, following Reviewer 2’s recommendation – see below), along with treatment success, and the within-subjects variable of time (baseline vs. follow-up). There were no significant two-way interactions involving PDOC and time (i.e. the changes in QOL between baseline and follow-up did not significantly differ between those with alcohol as PDOC and those with other drugs as PDOC). Moreover, there were no significant 3-way interactions between PDOC, treatment success, and time for physical, social, and environmental QOL, although for psychological QOL, we found that there was a significant 3-way interaction between PDOC, time, and treatment success (p=.028), but with small effect size (partial eta squared=.009). Examination of this interaction suggested that treatment success is more closely related to improved psychological well-being when alcohol is the PDOC compared to when the PDOC is an illicit drug. However, when testing the 2 PDOC groups separately, the interaction between time and treatment success was significant in both PDOC groups, so the main finding (that psychological QOL improves more in people who achieve treatment success than in those who don’t), remains consistent despite this significant 3-way interaction.

We have added the following paragraph to the results section of the main manuscript:

Further exploratory repeated measures ANOVA analyses examined whether these interactions were consistent across PDOC (i.e., alcohol versus illicit drug). There were no significant 3-way interactions between PDOC, time, and treatment success for environmental, social or physical QOL, but there was a significant 3-way interaction for psychological QOL (F(1,532)=4.874, p=.028, h2p=.009). This interaction is explored further in supplementary materials. Importantly, despite this 3-way interaction the 2-way interaction between time and treatment success was significant in both PDOC groups (see supplementary results), suggesting that the main finding (i.e. that improvement in psychological QOL was greater in treatment responders than in non-responders) was consistent, despite this association being stronger in those with alcohol as their PDOC. In addition, there were no 2-way interactions between PDOC and time, suggesting no general effect of PDOC on changes in QOL over time.”

In the supplementary results, we have added a more detailed description of this analysis, including an additional figure (Figure S2).

I would also like to see that other descriptive or demographic variables are included in future research

We have added the following sentence at the end of the discussion “Nonetheless, the influence of other variables (e.g. demographic and clinical characteristics) on changes in QOL, particularly whether other factors moderate or mediate associations between treatment success and improvement in quality of life, warrant examination in future research.”

Reviewer 2 Report

The authors utilized data collected in a large clinical sample from multiple sites and they examined the associations between QOL and substance-related treatment outcomes. The authors report, after substance abuse treatment, QOL improves in at least some domains. 

1) The Introduction is a bit too long and redundant.  This can be shortened to make it more succinct. 

2) The authors should compare the subjects included in the final analysis (N=536) with those who were not included due to lost follow up or missing data, to see if the differences.

3) Any differences in WHOQOL-BREF between male and female patients and among patients with different PDOC?  Since half of the patients reported alcohol as their PDOC, it might be worth running the analyses separately.  

4) Other clinically important factors such as comorbid psychiatric diagnosis was either not assessed or not included in the analysis. This needs to be mentioned as a limitation. 

Author Response

Reviewer 2.

The Introduction is a bit too long and redundant.  This can be shortened to make it more succinct

We have edited the introduction, shortening it further to accommodate some of the new material added in other sections as suggested by the reviewers.

The authors should compare the subjects included in the final analysis (N=536) with those who were not included due to lost follow up or missing data, to see if the differences.

We have analysed this, and added the following information to the results section:

“Participants missing from analyses due to either loss to follow-up or missing QOL or treatment outcome data (n=260) were significantly younger (mean=35.00±10.79 versus mean=38.16±10.64, t(789)=-3.897, p<.001). They were more likely to have amphetamines as their PDOC, and less likely to have alcohol as their PDOC (Pearson c2=12.784, p=.012). Participants recruited from residential rehabilitation (41.7%) and acute withdrawal (33.0%) were more likely to be missing from analyses than those recruited from outpatient settings (22%, c2=19.802, p<.001). There were no differences between those included in, and those missing from analyses, in gender (c2=1.553, p=.213) nor in any domain of baseline WHOQOL-BREF scores (all ps>.05).”

In addition, the paragraph in the discussion section regarding limitations, we have added the following sentence: “Moreover, participants missing from these analyses were younger than those included in these analyses, and differed from those included in terms of PDOC and PIT, and this may also bias our analyses and/or limit generalisability.”

Any differences in WHOQOL-BREF between male and female patients and among patients with different PDOC?  Since half of the patients reported alcohol as their PDOC, it might be worth running the analyses separately.

Regarding possible influences of PDOC on the findings presented, we have addressed this in response to Reviewer 1’s comments. We also note that in the main regression models, we controlled statistically for gender effects. Nonetheless, in response to the reviewer’s comment regarding gender, we conducted additional repeated measures ANOVAs to test whether gender moderated the interactions between treatment success and change in QOL. These analyses found no significant 3-way interaction between QOL change, treatment success, and gender on any QOL domain, so we have not added these analyses to the manuscript.

Other clinically important factors such as comorbid psychiatric diagnosis was either not assessed or not included in the analysis. This needs to be mentioned as a limitation.

Unfortunately we did not have a formal measure of psychiatric diagnosis and so have added this as a further limitation.